# The State of Education and Training for Antimicrobial Stewardship Programs in Indian Hospitals―A Qualitative and Quantitative Assessment

**DOI:** 10.3390/antibiotics8010011

**Published:** 2019-01-30

**Authors:** Sanjeev Singh, Esmita Charani, Chand Wattal, Anita Arora, Abi Jenkins, Dilip Nathwani

**Affiliations:** 1Amrita Institute of Medical Sciences, Amrita University, Kochi, Kerala 682041, India; 2NIHR Health Protection Unit in Healthcare Associated Infections and Antimicrobial Resistance, Imperial College London, London W12 OHS, UK; e.charani@imperial.ac.uk; 3Sir Ganga Ram Hospital, Rajinder Nagar, New Delhi 110061, India; chandwattal@gmail.com; 4Fortis Healthcare, Gurgaon Haryana 122001, India; docanitaarora@yahoo.com; 5British Society for Antimicrobial Chemotherapy, Birmingham B1 3NJ, UK; abijenkins@yahoo.co.uk; 6Ninewells Hospital and Medical School, Dundee 9SY, Scotland, UK; dilip.nathwani@nhs.net

**Keywords:** antibiotic stewardship, education and training, postgraduate, antimicrobial stewardship

## Abstract

*Background*: To understand the role of infrastructure, manpower, and education and training (E&T) in relation to Antimicrobial Stewardship (AMS) in Indian healthcare organizations. *Methods*: Mixed method approach using quantitative survey and qualitative interviews was applied. Through key informants, healthcare professionals from 69 hospitals (public & private) were invited to participate in online survey and follow up qualitative interviews. Thematic analysis was applied to identify the key emerging themes from the interviews. The survey data were analyzed using descriptive statistics. *Results*: 60 healthcare professionals from 51 hospitals responded to the survey. Eight doctors participated in semi-structured telephone interviews. 69% (27/39) of the respondents received E&T on AMS during undergraduate or postgraduate training. 88% (15/17) had not received any E&T at induction or during employment. In the qualitative interviews three key areas of concern were identified: (1) need for government level endorsement of AMS activities; (2) lack of AMS programs in hospitals; and, (3) lack of postgraduate E&T in AMS for staff. *Conclusion*: No structured provision of E&T for AMS currently exists in India. Stakeholder engagement is essential to the sustainable design and implementation of bespoke E&T for hospital AMS in India.

## 1. Introduction

Antimicrobial resistance (AMR) has risen steeply over the past few years and is now recognized as a global crisis. It is estimated that by 2050 AMR will kill ten million people at a cost of 100 trillion USD [1]. The World Health Organization (WHO) has identified AMR as an increasingly serious threat to global public health that requires action across all government sectors and society [2]. The Infectious Diseases Society of America (IDSA) has recognized antimicrobial stewardship (AMS) as a key intervention in combating AMR [3]. In India, global antimicrobial consumption is rising [4], and to date, the implementation and enforcement of the existing regulation on access to antibiotics has been difficult [4,5]. Antimicrobials continue to be dispensed without a prescription from accredited and non-accredited pharmacy outlets [6]. In 2012, the Chennai Declaration published recommendations for tackling AMR in India. The importance of education across the disciplines and key sectors were one of the top recommendations of this Declaration [7,8]. As the provision of education underpins good stewardship practice, understanding the availability, type and content of resources and the structures that support education are critical to effective implementation. There are no AMR surveillance and AMS education and training (E&T) data available from India [9]. Furthermore, a recent analysis of published stewardship activities in India reports a lack of good data to inform current and future practice and needs [10]. 

In India, E&T in healthcare in general has been largely fragmented, and static [11]. Traditionally, medical education in India has been heavily based on examinations. Postgraduate training is non-structured, non-competency based and varies between the medical colleges. In academic institutions, faculty development does not focus on new learning and there are opportunities for exploiting the power of information and technology to deliver teaching on AMS. Furthermore, individual competencies can be used as objective criteria to develop a set of values around the social accountability associated with antimicrobial prescribing [12,13]. The Indian healthcare system is very diverse, and application to be accredited by the National Accreditation Board for Hospitals (NABH) is not mandatory. For hospitals which voluntarily choose to undergo NABH accreditation, however, AMS is a mandatory indicator on which they are assessed. For hospitals choosing not to be accredited, there may be fewer incentives or drive to implement AMS or formalized E&T for health care staff on clinical parameters including AMS. In the absence of universal mandatory accreditation, the quality of care delivery varies and compliance to standard of care practices is only adhered to by a few institutions. This makes, implementation of uniform care services extremely challenging. 

Recognizing the need for more detailed information on the current state of E&T in AMS in Indian hospitals, the British Society for Antimicrobial Chemotherapy (BSAC) in 2014 in collaboration with a network of AMS leaders in India and the Science & Innovation Unit of the UK Foreign and Commonwealth Office (FCO) in India, established a project aiming to formulate a national framework for locally relevant training that will support AMS activities in Indian hospitals. To do this, quantitative and qualitative studies were undertaken aiming to understand the role of education, training, infrastructure, and manpower in relation to AMS in Indian healthcare organizations. In this paper, we report the findings of these studies. 

## 2. Materials and Methods

Retrospective Institutional Ethical Committee approval was granted by the lead study site in India (Amrita Institute of Medical Sciences: AIMS/IEC/2017/347).

A mixed methodology was adopted with quantitative survey of healthcare professionals working in Indian hospitals followed up by qualitative semi-structured telephone interviews with AMS leads in hospitals across India. Institutions which had taken steps towards organizing and implementing AMS were purposively selected for participation in this study. Participation in the studies was voluntary and the respondents were not mandated to participate or to respond to all the questions. Informed consent was taken from each respondent prior to data collection.

### 2.1. Quantitative Survey

Using purposive sampling, sixty-nine healthcare organizations in the private (54) and public sector (15) were identified by the study collaborators in India (Sanjeev Singh and Chand Wattal). Expert physicians with a role in AMS in each organization were invited to participate in the survey. Purposive sampling enabled the researchers to target individuals from organizations which were known to have initiated some AMS efforts.

An internet-based survey (available as Appendix A), based on the study objectives, was developed by the study collaborators at the end of 2015. The survey was piloted in five centers identified by the collaborative network in December 2015, and was subsequently revised in response to the feedback received. The survey included Likert scale questions asking the respondents to rate which components they considered as important for inclusion in E&T for AMS. The survey was then circulated electronically using Survey Monkey to the identified individuals during January and February 2016. Participants were assured of anonymity when agreeing to participate in the survey. The survey asked questions relating to current and preferred E&T practices about AMS/antimicrobial prescribing as an individual practitioner and were asked to mark their views on Likert scale (1–5).

### 2.2. Qualitative Interviews

Twenty-five of the survey participants were purposively selected to represent both public and private sector institutions and were invited to participate in qualitative interviews. Individuals were selected for having responsibility in AMS E&T. A semi-structured interview guide was developed by E.C. based on ongoing research on the implementation of AMS in different healthcare organizations. The interview guide was further edited and adapted for this study in India through interviews and correspondence with the study collaborators. The interview topics included questions on existing AMS activities in the organization; E&T on AMS provided to the participant; any E&T activities the participant is involved in as part of their job; what level of resources, both human and material, were available to dedicate to E&T and AMS. Open ended questions were used as prompts to explore the participants’ views, perceptions and experiences of the status of education and training in, and implementation of AMS in Indian hospitals. The interview guide (available as Appendix A) was piloted prior to the study. An email invitation (which included the study information leaflet) to participate telephone interviews for this study was sent to the twenty-five individuals. The initial email was sent by S.S. and C.W., and a follow up reminder email was sent to those who had not responded after two weeks. All the interviews were conducted on a day and time convenient for the participants. Written consent to participate in the study was obtained from participants by email. The interviews were conducted in English. All the telephone interviews were audio recorded. The interviews were then transcribed verbatim. Prior to analysis all participant identifying data was removed from the transcripts. 

### 2.3. Analysis: Quantitative Survey and Qualitative Interviews

The quantitative survey data was compiled and collated for E&T at undergraduate and postgraduate level as well as in current employment, participant role regarding AMS, induction training on AMS, AMS components covered and relevance of each one of the components and preferred methods of training.

The data from the qualitative interviews was analyzed using NVivo10 qualitative analysis software (QSR International, USA). The analysis was conducted in an iterative manner, using a constant comparative process to identify the emerging key themes from the transcripts. The transcripts were at first open coded, the codes were then iteratively grouped to develop the thematic framework [14]. Interviews were conducted until saturation of the emerging themes were achieved. Saturation in this context meant that no new themes were emerging from the data. Using a single researcher (EC) to conduct the interviews and the core analysis facilitated an in-depth analysis of the data and the emerging themes. The emerging themes and findings were discussed with the co-authors to reach consensus.

## 3. Results

### 3.1. Quantitative Survey

Of the 69 hospitals invited, 60 (74%) completed the survey. Of the respondents, 49/60 (82%) represented faculty from private institutions and 11/60 (18%) for public institutions (Table 1). Undergraduate E&T in antimicrobial prescribing was reported by 39/60 (65%) while postgraduate E&T in antimicrobial prescribing was reported by 35/60 (58%) of respondents. 19/60 (32%) of the respondents reported not receiving any E&T during undergraduate training and 16/60 (27%) of the respondents reported not receiving any E&T during post-graduate training. 

Of those who responded, 28/49 (57%) of private hospitals and 4/11 (36%) of government hospitals provide E&T in AMS and or infection control at induction to their staff. The principal source of this postgraduate training was from an affiliated university or college whereas other sources for providing training such as professional societies or governmental agencies were few.

In response to the Likert questions on the AMS components that should be covered within a potential E&T curriculum all components (including: (1) minimizing unnecessary prescribing of antimicrobials; (2) adopting necessary infection control & prevention measures; and, (3) obtaining appropriate microbiological samples for culture & sensitivity tests) scored equally highly.

The questions on methods for the delivery of E&T revealed a preference for more interactive learning based around real world data (bedside teaching linked to their clinical practice) and case scenarios. There was support for providing this in conjunction with an e-learning resource - the hybrid learning approach as opposed to e-learning only (Table 2). Furthermore, linking the education into continuing medical education and formal certification/accreditation was suggested as a useful approach.

### 3.2. Qualitative Semi-Structured Interviews

Of the 25 individuals invited, 20 agreed to participate in the study. Of these twenty we were able to interview eight participants. The remaining twelve though they had agreed, either dropped out or we were unsuccessful in establishing contact for telephone interviews. The participants interviewed were microbiologists, anesthetists, and intensive care physicians (interview participant data available as Appendix A) Though the invitations were sent to clinicians in both private and public hospitals, the response rate was mainly from private hospitals, with one participant working in a public hospital. Saturation, whereby no new themes in relation to E&T were emerging from the participant responses, was reached in the interviews from the private hospitals. In relation to E&T in AMS the interviews highlighted three key areas: (1) the need for government level endorsement and governance of AMS; (2) lack of structure in teaching about AMR and AMS at undergraduate and post-graduate level; and, (3) lack of AMS programs in hospitals.

#### 3.2.1. The Need for Government Level Endorsement and Governance of AMS 

There is a general increase in awareness and willingness to implement AMS activities in hospitals, however the participants reported an acute awareness of the need for there to be government level initiatives, support and legislation.
We want to get it going, but there’s a lot of angst at that. If you talk to somebody, they’re not very open to taking suggestions. And from the governance mechanisms we don’t have anything.Medical microbiologist

At hospitals, initiatives to support AMS are being implemented, however the participants report that adoption and adherence to these initiatives would be better if there was a central mandatory impetus to support the local initiatives (Table 3, T1–T4).

#### 3.2.2. Lack of Structure in Teaching about AMR and AMS at Undergraduate and Post-Graduate Level 

Education and training for AMS is not formalized or mandatory in the undergraduate or the post-graduate curriculum, leaving individuals to form habits that are difficult to change (Table 3, T5–T8).
What happens is we end up doing, and I’m being very frank with you here, we end up learning things which are so difficult to unlearn later. So even though you may have resources at your disposal later, what you have learned over the years, it becomes difficult to unlearn that and learn new things. And in our medical curriculum, infection control, antibiotic policies, they are not hammered so to speak. Maybe just a byline or a small chapter somewhere. Enough focus is not being given to them.Medical microbiologist

This means that the non-specialist doctors have little knowledge and understanding of antimicrobial prescribing and AMS (below quote and see also Table 3, T8).
Other than microbiologists, in general, the doctors are not much aware about this problem (referring to AMR), the drug resistance and such things.Medical microbiologist

#### 3.2.3. Lack of AMS Programs in Hospitals 

There are no mandatory AMS programs in Indian hospitals. Though the awareness of the need for stewardship is increasing, the current training provided is mainly by microbiology departments and attendance is based on individual doctors’ incentives (below and Table 3, T9).


*We have a weekly meeting where in generally there is a small capsule about 15 min on a drug or a group of drugs which is attended by a lot of the consultants of the hospital, now this is not very structured teaching but in so much as sharing information, a microbiologist along with the pharmacist actually discuss one more drug in the meeting, so there a lot of people who don’t really understand a lot of the pharmacology, and they’re all like 55, 60 plus now so there are a lot of consultants who are ageing so there is a focus on teaching and training them, the other thing that we’re trying to also do is the clinical pharmacist round. It normally does a fair amount of discussion, the clinical pharmacist acts like, the idiot in the group and actually says I know you recommended this but I do not understand so could you just tell me how you do this and what’s the rationale so if anything in a kind of a dialogue and that’s something which is found to be pretty effective.*
Medical microbiologist

This gap in structure is in some places being addressed by interventions developed by pharma (Table 3, T10). There is an opportunity for developing hybrid learning which includes face to face discussion with interactive e-learning modules to help support doctors in implementing AMS and optimizing their antibiotic prescribing (Table 3, T11–12).

## 4. Discussion

The findings in this paper highlight gaps in: (1) AMS provision in secondary/tertiary care system in India; and, (2) the provision of education and training for AMS. One third of respondents in this study report as receiving no training in AMS, with some believing that this would not be relevant to their current practice. More than half of respondents reported postgraduate E&T in AMS, but one third reported that their employing organization provided this during employment. There is a disparity between these two findings. Possible reasons for such a difference could be explained if the respondents do not see this employer-provided education as relevant in this case, they are not partaking in their institutional education and training, or they consider they are not receiving this since they are the ones providing it.

The qualitative interviews highlighted that there are currently no mandatory programs at national or state level to implement AMS or provide training for AMS. The respondents unanimously reported that one solution to improving uptake of AMS programs and training is to bring in governance and nationally mandatory programs. Active participation in teaching and policy development is an important characteristic of organizations that support training of staff [11,12] Most of the respondents in this study had a role in teaching and policy development. Interestingly, it is this teaching group that appeared to be the least likely to have received undergraduate or post graduate E&T in AMS. This suggests a gap and potential opportunity for development of a ‘training the trainers’ module or course to support educators to deliver robust, high quality, evidence-based and up-to-date AMS education and training [13,15]. The value of such courses and how they may be constructed to reflect the required skills sets and competencies of healthcare work force has been outlined in the literature [16]. Using this information to retain healthcare professionals and to construct a bespoke curriculum that meets the needs of those planning and involved in delivering training and is sensitive to local structures, resources and culture is key towards implementation of AMS. Attention needs to be paid to the content, volume, quality and resources available to deliver AMS in a cost-effective manner [17,18]. Due to the lack of presence or access to good structured and consistent education and training programs, pharmaceutical companies have stepped in, in some hospitals, to provide training to staff. It is essential to have robust training and understanding of application of AMS at undergraduate and postgraduate level for appropriate prescription practices for patient safety.

The delivery of training that is resource efficient and cost-effective is a key consideration. Amongst the respondents to the qualitative interviews there was an overall positive response to providing interactive training via technology, such as smartphones and e-learning [19]. These are opportunities that can be explored as part of new AMS programs for Indian hospitals. Currently the primary mode of delivery of training in AMS is via face-to-face presentations. This, however, has the lowest preference for method of delivery by the participants in study. Preferred methods were those that are more interactive such as ‘on the job cased based’ training in addition to using a combination of face-based teaching complimented by e-learning resources. This type of hybrid approach to training is one that is being increasingly used in AMS programs [11,20].

A review of the content of AMS E&T both at induction and throughout employment demonstrates some gaps in the breadth of topics covered. Infection control and prevention is comprehensively covered in the existing E&T provided by healthcare organizations, however, core components of stewardship training are not well covered. This presents an opportunity for development, as AMS practices should be integral to good infection prevention and control [9]. For example, good practice in both these areas can reduce the occurrences of line-related complications and facilitate hospital discharge also affecting the cost of care.

One of the secondary objectives of our survey was to get some information about infra-structure to support infection prevention and stewardship. It is encouraging that all respondents confirmed that there was an ‘infection control and prevention’ and/or ‘antimicrobial stewardship’ committee already established within their healthcare organization. There is a need for having committees, top level governmental and institutional, to oversee AMS work. This need was also identified and is consistent with the recommendations of the Chennai declaration [8]. Furthermore, our findings suggest that these structures have produced outputs such as organizational policies and restricted antimicrobial lists. 

We recognize that this study has some limitations. There are inherent flaws with web-based surveys of this kind. There is also sample selection bias, the low response rate in view of the large hospital population in India and the low participation from public hospitals. Co-ordination of this survey was challenging, and getting access to those responsible for running such programs often required personal requests for the India-based researchers involved in this study. These factors and others perhaps illustrate why so few studies exist from Indian hospitals. Despite this, our study represents what we believe to be the largest published study of its kind, with a focus on education and training for AMS in India. Another limitation is the lack of representativeness from government healthcare organizations in the qualitative interviews. The data from the qualitative interviews is primarily from private healthcare organizations. We found access to healthcare staff in the government hospitals more challenging, due in part to the processes required for them to be involved in research and also due to the time constraints and work load which prohibits them from being actively involved such research.

## 5. Conclusions

The results of this study demonstrate the inconsistent and fragmented nature of E&T provision for antimicrobial prescribing and AMS in India. There is an unmet need for AMS activity in India, but a common infrastructure, or formal/mandatory framework, to support and legislate for the provision of AMS and E&T, is lacking. More work is needed to ascertain the training needs of those working in public and governmental health organizations and for the broader multi-professional health care team. There is a strong recognition for the need for mandatory policies and governance on AMS to guide the local implementation of programs in hospitals.

## Figures and Tables

**Table 1 antibiotics-08-00011-t001:** Employer provided education and training (E&T) at induction and throughout employment.

Do Healthcare Workers at Your Institution or Organization Receive E + T in Antimicrobial Stewardship and/or Infection Control at Induction (within Three Months of Starting Their Job)?	No. Responses		Have you Received Postgraduate Training in Antimicrobial Prescribing?	No. Responses *n* (%)		Have you Received Undergraduate Training in Antimicrobial Prescribing?	No. Responses *n* (%)	
	Private Hospitals	Public Hospitals		Private Hospitals	Public Hospitals		Private Hospitals	Public Hospitals
Yes	28 (57)	4 (36)	Yes	16 (33)	3 (27)	Yes	33 (65)	6 (55)
No	17 (35)	7 (64)	No	26 (53)	7 (64)	No	14 (28)	5 (46)
Not Sure	4 (8)	0	Not Sure	7 (14)	1 (9)	Not Sure	2 (4)	0
Total	49 (100)	11 (100)	Total	49 (100)	11 (100)	Total	49 (100)	11 (100)

**Table 2 antibiotics-08-00011-t002:** Preferred methods of E&T for Antimicrobial Stewardship (AMS). Likert score range was 1 (Least preferred) to 5 (highly preferred).

E&T Methods	No of Respondents		Mean Likert Score
	Induction	Follow-Up	
Face-to-face lectures or presentation	25	15	3.87
Face-to-face workshops or seminars	10	9	4.31
Work-place teaching e.g., workbooks or portfolios	10	9	4.28
On the job’ learning from practice	19	12	4.45
Web-based or e-learning	2	2	3.64
Mixed Methods (face to face interview + E learning)	7	4	4.40
**Total**	**28**	**17**	**51**

**Table 3 antibiotics-08-00011-t003:** The key emerging themes on education and training in antimicrobial resistance (AMR) and AMS from the semi-structured interviews.

Themes	Quotes
The need for government level endorsement and governance	*T1* *We want to get it going, but there’s a lot of angst at that. If you talk to somebody, they’re not very open to taking suggestions. And from the governance mechanisms we don’t have anything.* Medical microbiologist
	*T2* *Actually, at this point in time we don’t have any restrictions. We’re going to plan it soon, after tomorrow’s program. The Government is trying to bring in antibiotic policy for all hospitals in the public and the private sector.* Medical microbiologist
	*T3* *On a scale of zero to ten, where ten is appropriate antibiotic usage, we are at about two or three. It’s mainly because of ignorance. Second, a lack of federal laws preventing over the counter dispensation of antibiotics. And third is a lack of knowledge about how antibiotics work and how their patients might not benefit by abuse of antibiotics. So it boils down to law and education.* Anaesthetist, Chair of infection prevention and control
	*T4* *In an ideal world I would like an online module where it is a requirement that you go onto it, you have got to pass it. That would be a good way if it is a mandate or if it’s compulsory and they need to do it.* Medical microbiologist
Lack of structure in the current education and training efforts in AMS	*T5* *I**n India, what the system is in a medical school, it’s in our second year probably you’re taught about pharmacology. And the clinical rounds start from the third year. So by the time you start your clinical rounds, it’s a very bookish language, and how to interpret it clinically is not something which is really taught. But when it comes to prescribing it’s more like what you see around. Your seniors doing it, your colleagues doing it. Not at the undergrad level but the post grad level, what the medical representatives are coming and talking to you about. So there are no structured programs talking about these antibiotic prescriptions.* Medical microbiologist
	*T6* *We definitely lack good formal education in this. Both undergraduate as well as post graduates. Treatment is largely taught, but stewardship is still not a part of the curricula. All they read is Harrison Textbook of Medicine. That tells you beautifully about how to treat the patient. Unfortunately, it cannot teach you when not to prescribe.* Anaesthetist, Head of ICU
	*T7* *I don’t think people are willing to put enough structure to any program, a lot of doctors work in an unstructured way it is not yet come on the curriculum it’s not seen as a part and parcel of clinical practice training and I think it should be there.* Medical microbiologist
	*T8* *I personally feel that antibiotic prescribing and infection control is not a priority still today in the medical curriculum. And that produces a huge gap in the training issues. Most of my residents initially when we take them on, they have no idea about what antibiotic I’m talking about. What bacteria I’m talking about. They’ve heard about the name.* Anaesthetist, Chair of infection prevention and control
Lack of AMS programs in Indian hospitals	*T9* *We did a project on antimicrobial stewardship, the surgical prophylaxis we took it as a project and then under this stewardship program we do a lot of education classes with our team. So, these have at least a once a month session going for about a year or so, and we would audit it every quarter. So after one intervention we audited, we give a feedback. Initially we were doing a monthly audit on the prescription practices for surgical prophylaxis only, but now we do it on a once a quarter.* Medical microbiologist
	*T10* *Actually a lot of pharma companies have developed their e-learning sites which are not always biased. But over time I’ve realized that busy doctors usually do not visit e-learning places. Still in my state unfortunately, we organize workshops, we do give credit points as per the Medical Council, but it’s not mandatory. I have serious doubts whether e-learning would help. We can have a classroom style thing which, presented interestingly, people are interested.* Medical microbiologist
	*T11* *I think it is always better to provide the teaching face-to-face. It provides them a platform to ask questions, get real-time feedback, any inhibitions or any confusion they have regarding what is communicated, they can sort it out and we get much better buy-in. The online education programs are good for people with an interest and they are into e-learning. So if I am interested in learning something, I would be willing to go through an online training program and clear it. It’s very good for me because that buy-in is already there. But if you’re talking to a group of people whom you want to convert or move to your side, whom you want to change their behavior, you want to change their outlook, in that aspect always face-to-face mediation would fare better.* Medical microbiologist
	*T12* *Number one, number two and number three is e-education. If we can get any help in e-education and assessment. That’s all I would want. I don’t want anything else. Because everybody in India has a smartphone. They can use a smartphone to access your website and answer a survey or go through a particular brochure, guidelines. And second thing is we can tag their appraisal to passing of these e-tutorials. So there is no pressure, but there is pressure. It’s all about education, e-learning, e-assessment and sharing of data or making some kind of groups where people share their success stories and their failures.* Anaesthetist, Chair of infection prevention and control

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
