# Peer review of "The State of Education and Training for Antimicrobial Stewardship Programs in Indian Hospitals―A Qualitative and Quantitative Assessment"

_antibiotics, 2019, doi:10.3390/antibiotics8010011_

Round 1

Reviewer 1 Report

General comments

This paper addresses an important issue of improving antimicrobial stewardship at hospitals in India where state-based AMS effort does not exist. The authors combine quantitative and qualitative methodologies to understand the current states of AMS education and training efforts by individual hospitals and opportunities and challenges to improve it. While the paper has potential to contribute to the literature, there are some questions about the structure of the paper, sample selection, and presentation of the results. Detailed suggestions and recommendations are provided below.

Introduction

-       I find it very hard to believe that there is “no regulation on access to antibiotics” in India. A quick literature search yielded the below studies that discuss challenges of enforcing existing policies, rather than the lack of them. The authors may review such evidence to communicate a more nuanced picture regarding India’s effort to control AMR and irrational antibiotic use.

https://www.ncbi.nlm.nih.gov/pmc/articles/PMC3193708/

https://journals.plos.org/plosmedicine/article?id=10.1371/journal.pmed.1001974

http://apps.who.int/medicinedocs/en/m/abstract/Js20299en/

-       The introduction is rather contradictory. On page 2, authors state that there is no evidence regarding AMR surveillance and AMS education and training. However, the second paragraph on page 2 starts by saying that “E&T in healthcare has been largely fragmented and static”, suggesting that some evidence exists.

-       The link between accreditation by NABH and AMS E&T is unclear and needs to be explained. Are the authors implying that the lack of NABH accredited institutions lead to limited AMS E&T programs available throughout the country? If so, why and how? Or do they wish to communicate that there is no standardized quality assessment criteria to develop AMS E&T?

-       From the introduction, it is unclear what the authors mean by AMS. Initially I understood that stewardship refers to activities at medical institutions, but, if I understood it correctly, the quantitative survey addresses E&T prior to participants joining their respective medical institutions. I am uncertain how participants’ prior E&T relates to hospital-level stewardship. This needs to be better explained.

Materials and methods

-       It would be good if the authors could explain what “purposive sampling” meant in their context. They described that a handful of hospitals were identified by the study collaborators, central government sources, local professional groups, and the British Foreign and Commonwealth Office, but how was this identification done, what criteria were used by them, and why? Is this part of the reason why many more private hospital were selected compared to public ones? Have the authors selected secondary and tertiary hospitals on purpose? Also, is a physician always responsible for AMS at an organization, or did the authors deliberately select institutions where a physician is in charge of AMS? If so, why?

-       It is unclear to me if participants were asked about AMS E&T offered during their employment at the institutions they work at as well as AMR-related E&T prior to joining their institutions. At the end of Section 2.3, the authors described the topics of the semi-structured interviews as “existing AMS activities in the organization; post-graduate E&T on AMS provided to the participant; any E&T activities the participant is involved in as part of their job; what level of resources, both human and material, were available to dedicate to E&T and AMS.” However, in Section 2.3, the authors indicate that participants were asked to answer questions regarding training at the undergraduate and postgraduate levels. Also, by “postgraduate level” do the authors mean training on the job (i.e. at the institutions employing participants) or MSc or equivalent that participants undertook away from the institutions they are employed at? These points need to be clarified.

Results

-       Table 2 needs to be made clearer to convey key messages. First, the paper does not explain what the scale of the Likert score was (e.g. between 0 and 5, etc.). Without this information, it is difficult to interpret the mean scores. Also, the mean Likert score seems to be relatively similar, indicating that it is difficult to demonstrate a clear hierarchy in terms of what participants considered as important. Would it be possible to conduct a t-test or equivalent to see if there is a statistically significant difference in mean values? Alternatively, the authors could present variances to see if there is much variation in opinions expressed by the participants. Finally, it would be good to indicate why the number of respondents differ between induction and follow-up E&T.

-       Table 3 does not make it clear what responses correspond to current and preferred methods of E&T.

-       How were eight participants selected out of the 20 invited individuals for the qualitative semi-structured interviews?

-       The quotes presented are generally supportive of the authors’ claim. However, they are rather lengthy in the current form and perhaps it would be better to further extract parts of the quotes that emphasize the authors’ key messages. Also, the total number of direct quotes may be reduced.

-       As said earlier, the link between undergraduate/post-graduate level AMS E&T and hospital-level AMS E&T needs to be made clear. It seems that the authors are arguing that a sound basis and awareness of AMR needs to be cultivated while medical professionals are still students. But this message does not come across clearly in the current form and can be improved.

Discussion and Conclusions

-       The discussion section returns to the question posed in the introduction, but introduces new insights that were not discussed in the Results section. For instance, the authors refer to infection control and prevention and/or antimicrobial stewardship committees at medical institutions. It would be good to see this data in the Results section to better understand the fragmented nature of the existing AMS E&T programs.

-       It would have been good to see more description of the participants in earlier sections. The second paragraph of the Discussion section brings up an interesting point about participants who were in the “teaching group” were least likely to have received UG or PG education. In the current form, the paper does not adequately present characteristics of the participants. This can be included in the earlier section (e.g. Materials and methods, or Results).

Author Response

We thank the reviewer for their comments on this manuscript. We provide below a point by point response. 

Reviewer 1:

General comments

This paper addresses an important issue of improving antimicrobial stewardship at hospitals in India where state-based AMS effort does not exist. The authors combine quantitative and qualitative methodologies to understand the current states of AMS education and training efforts by individual hospitals and opportunities and challenges to improve it. While the paper has potential to contribute to the literature, there are some questions about the structure of the paper, sample selection, and presentation of the results. Detailed suggestions and recommendations are provided below.

Response

We are grateful for this reviewer recognising the importance of this study from India and provide response to the suggestions below.

Comment

Introduction

-       I find it very hard to believe that there is “no regulation on access to antibiotics” in India. A quick literature search yielded the below studies that discuss challenges of enforcing existing policies, rather than the lack of them. The authors may review such evidence to communicate a more nuanced picture regarding India’s effort to control AMR and irrational antibiotic use.

https://www.ncbi.nlm.nih.gov/pmc/articles/PMC3193708/

https://journals.plos.org/plosmedicine/article?id=10.1371/journal.pmed.1001974

http://apps.who.int/medicinedocs/en/m/abstract/Js20299en/

Response

We have amended the text to the following, and added suggested references.

‘In India, global antimicrobial consumption is rising [2],and to date, the implementation and enforcement of the existing regulation on access to antibiotics has been difficult [2,3]. Antimicrobials continue to be dispensed without a prescription from accredited and non-accredited pharmacy outlets.’

Comment

-       The introduction is rather contradictory. On page 2, authors state that there is no evidence regarding AMR surveillance and AMS education and training. However, the second paragraph on page 2 starts by saying that “E&T in healthcare has been largely fragmented and static”, suggesting that some evidence exists.

Response

The second sentence refers to date on E&T generally and not specific to AMR surveillance and AMS. We have clarified this further in the text to:

‘In India, E&T in healthcare in general has been largely fragmented, and static[11]. Traditionally medical education in India has been heavily based on examinations.’

Comment

-       The link between accreditation by NABH and AMS E&T is unclear and needs to be explained. Are the authors implying that the lack of NABH accredited institutions lead to limited AMS E&T programs available throughout the country? If so, why and how? Or do they wish to communicate that there is no standardized quality assessment criteria to develop AMS E&T?

Response

We have edited the sentence to clarify further:

‘The Indian healthcare system is very diverse, and application to be accredited by the National Accreditation Board for Hospitals (NABH) is not mandatory. For hospitals which voluntarily choose to undergo NABH accreditation, however, AMS is a mandatory indicator on which they are assessed. For hospitals choosing not to be accredited, there may be fewer incentives or drive to implement AMS or formalized E&T for health care staff on clinical parameters including AMS. In the absence of universal mandatory accreditation, the quality of care delivery varies and compliance to standard of care practices is only adhered to by a few institutions. This makes, implementation of uniform care services extremely challenging.’

Comment

-       From the introduction, it is unclear what the authors mean by AMS. Initially I understood that stewardship refers to activities at medical institutions, but, if I understood it correctly, the quantitative survey addresses E&T prior to participants joining their respective medical institutions. I am uncertain how participants’ prior E&T relates to hospital-level stewardship. This needs to be better explained.

Response

Our study had two purposes. First to understand E & T with regards to AMS received at the time of joining the hospital they are working in and secondly if they received any AMS E&T during their undergraduate studies.

Comment

Materials and methods

-       It would be good if the authors could explain what “purposive sampling” meant in their context. They described that a handful of hospitals were identified by the study collaborators, central government sources, local professional groups, and the British Foreign and Commonwealth Office, but how was this identification done, what criteria were used by them, and why? Is this part of the reason why many more private hospital were selected compared to public ones? Have the authors selected secondary and tertiary hospitals on purpose? Also, is a physician always responsible for AMS at an organization, or did the authors deliberately select institutions where a physician is in charge of AMS? If so, why?

Response

We have revised the text to below:

‘Using purposive sampling, sixty nine healthcare organizations in the private (54) and public sector (15) were identified by the study collaborators in India (SS and CW). Expert physicians with a role in AMS in each organization were invited to participate in the survey. Purposive sampling enabled the researchers to target individuals from organizations which were known to have initiated some AMS efforts.’ 

The fact that government hospitals are underrepresented in this sample is indicative of the limited AMS activities in the public sector, due to numerous resource limitations. This is later picked up in the discussion.

Comment

-       It is unclear to me if participants were asked about AMS E&T offered during their employment at the institutions they work at as well as AMR-related E&T prior to joining their institutions. At the end of Section 2.3, the authors described the topics of the semi-structured interviews as “existing AMS activities in the organization; post-graduate E&T on AMS provided to the participant; any E&T activities the participant is involved in as part of their job; what level of resources, both human and material, were available to dedicate to E&T and AMS.” However, in Section 2.3, the authors indicate that participants were asked to answer questions regarding training at the undergraduate and postgraduate levels. Also, by “postgraduate level” do the authors mean training on the job (i.e. at the institutions employing participants) or MSc or equivalent that participants undertook away from the institutions they are employed at? These points need to be clarified.

Response

We wanted to capture both any training the participants may have received as part of their undergraduate training or post-graduate training (e.g. self-attended courses, further qualifications etc) in addition to any training provided by current employers. We asked these separately to be able to capture any opportunities the participants have had for training and education in AMS.

Comment

-       Table 2 needs to be made clearer to convey key messages. First, the paper does not explain what the scale of the Likert score was (e.g. between 0 and 5, etc.). Without this information, it is difficult to interpret the mean scores. Also, the mean Likert score seems to be relatively similar, indicating that it is difficult to demonstrate a clear hierarchy in terms of what participants considered as important. Would it be possible to conduct a t-test or equivalent to see if there is a statistically significant difference in mean values? Alternatively, the authors could present variances to see if there is much variation in opinions expressed by the participants. Finally, it would be good to indicate why the number of respondents differ between induction and follow-up E&T.

Response

The legend for table now includes the following sentence:

‘Table 2. AMS Topics Covered in E&T and Importance Attributed to Each. Likert score range was 1 (not important) to 5 (very important)’

We agree this table may not be adding much to the narrative of the paper and have removed the table, adding the following to the results section: In response to the Likert questions on the AMS components that should be covered within a potential E&T curriculum all components (including minimizing unnecessary prescribing of antimicrobials; 2) adopting necessary infection control & prevention measures; and, 3) obtaining appropriate microbiological samples for culture & sensitivity tests) scored equally highly.

Comment

-       Table 3 does not make it clear what responses correspond to current and preferred methods of E&T.

Response

This was a typo and has been now corrected to the following:

‘Table 3. Preferred methods of E&T for AMS. Likert score range was 1 (not important) to 5 (very important)’

Comment

-       How were eight participants selected out of the 20 invited individuals for the qualitative semi-structured interviews?

Response

We have added text to clarify this:

‘Of the 25 invited individuals, 20 accepted to participate in the study. Of these twenty we were able to interview eight participants. The remaining twelve though they had agreed, either dropped out or we were unsuccessful in establishing contact for telephone interviews. The participants interviewed were microbiologists, anaesthetists, and intensive care physicians (interview participant data available as supplementary files).’

Comment

-       The quotes presented are generally supportive of the authors’ claim. However, they are rather lengthy in the current form and perhaps it would be better to further extract parts of the quotes that emphasize the authors’ key messages. Also, the total number of direct quotes may be reduced.

Response

Where possible we have reduced the quotes without taking away the contextual meaning. These are all track changed in the manuscript.

Comment

-       As said earlier, the link between undergraduate/post-graduate level AMS E&T and hospital-level AMS E&T needs to be made clear. It seems that the authors are arguing that a sound basis and awareness of AMR needs to be cultivated while medical professionals are still students. But this message does not come across clearly in the current form and can be improved.

Response

It is essential to have robust training and understanding of application of AMS at undergraduate and postgraduate for appropriate prescription practices for patient safety. Has been added from 252-254.

Comment

-       The discussion section returns to the question posed in the introduction, but introduces new insights that were not discussed in the Results section. For instance, the authors refer to infection control and prevention and/or antimicrobial stewardship committees at medical institutions. It would be good to see this data in the Results section to better understand the fragmented nature of the existing AMS E&T programs.

Response

This has been covered in Table 1, wherein healthcare workers at institutional level have been asked whether AMS and / or infection control at induction has been covered. The response has been No: 17 (35) in private hospitals and 7 (64) in public hospitals.

Comment

-       It would have been good to see more description of the participants in earlier sections. The second paragraph of the Discussion section brings up an interesting point about participants who were in the “teaching group” were least likely to have received UG or PG education. In the current form, the paper does not adequately present characteristics of the participants. This can be included in the earlier section (e.g. Materials and methods, or Results).

Response

We hope the changes made has now clarified this.

Reviewer 2 Report

This manuscript describes a survey made in 69 hospitals throughout India to establish the state of training (undergraduate and postgraduate) in antimicrobial stewardship in the country. The eye-opening results are both quantitative and qualitative and give orientation for AMS leaders in India for future battles. I would recommend this manuscript for publication with minor edits:

Tables could benefit to have a legend and a small description.

I didn’t feel Table 1 was necessary. Most of the information can be digested in the text.

In Tables 2 and 3, the added total responders is in the Likert Score column and should be put elsewhere or omitted

While mentioned in the discussion, distinction between induction and follow-up E&T should be more clearly stated and should be done earlier in the manuscript (methods would be best)

In the methods, the range of the Likert Scale should be mentioned. I would also appreciate some kind of dispersion measures instead of just a mean.

In the results (line 161), no underlined subtitle to introduce The need for government level endorsement and governance of AMS

In the discussion (line 224),UG and PG abbreviations are used without being explicited earlier. Being introduced late in the text, I would suggest to write the unabbreviated words.

While the introduction and discussion are well written, they could be slightly shortened to help readers to get major points.

Author Response

Reviewer 2

Comment

This manuscript describes a survey made in 69 hospitals throughout India to establish the state of training (undergraduate and postgraduate) in antimicrobial stewardship in the country. The eye-opening results are both quantitative and qualitative and give orientation for AMS leaders in India for future battles. I would recommend this manuscript for publication with minor edits:

Response

Thank you for recognising the importance of this work to the context in India.

Comment

Tables could benefit to have a legend and a small description.

I didn’t feel Table 1 was necessary. Most of the information can be digested in the text.

Response

We have added brief descriptions to table as per advice of both reviewers. We will leave it to the discretion of the editors if they think it appropriate to remove table 1.

Comment

In Tables 2 and 3, the added total responders is in the Likert Score column and should be put elsewhere or omitted

Response

We agree with this comment. We have removed table 2, and have added the Likert Score to the existing table (now table 2).

Comment

While mentioned in the discussion, distinction between induction and follow-up E&T should be more clearly stated and should be done earlier in the manuscript (methods would be best)

Response

Comment

In the methods, the range of the Likert Scale should be mentioned. I would also appreciate some kind of dispersion measures instead of just a mean.

Response

Added in Methods section: Line 100; antimicrobial prescribing as an individual practitioner and were asked to mark their views on Likert Scale (1-5).

Comment

In the results (line 161), no underlined subtitle to introduce The need for government level endorsement and governance of AMS

Response

We have added this subtitle.

Comment

In the discussion (line 224),UG and PG abbreviations are used without being explicited earlier. Being introduced late in the text, I would suggest to write the unabbreviated words.

Response

We have edited the text to include the unabbreviated words.

Comment

While the introduction and discussion are well written, they could be slightly shortened to help readers to get major points.

Response

Thank you. We hope the editions made have made these sections more clear.
